# Accurate Cold-start Bundle Recommendation via Popularity-based Coalescence and Curriculum Heating

## ABSTRACT

How can we accurately recommend cold-start bundles to users? The cold-start problem in bundle recommendation is crucial in practical scenarios since new bundles are continuously created on the Web for various marketing purposes. Despite its importance, existing methods for cold-start item recommendation are not readily applicable to bundles. They depend overly on historical information, even for less popular bundles, failing to address the primary challenge of the highly skewed distribution of bundle interactions. In this work, we propose CoHEAT (Popularity-based Coalescence and Curriculum Heating), an accurate approach for cold-start bundle recommendation. CoHEAT first represents users and bundles through graph-based views, capturing collaborative information effectively. It then tackles the highly skewed distribution of bundle interactions by incorporating both historical and affiliation information based on the bundle's popularity when estimating the user-bundle relationship. Furthermore, it effectively learns latent representations by exploiting curriculum learning and contrastive learning. CoHEAT demonstrates superior performance in cold-start bundle recommendation, achieving up to 193% higher nDCG@20 compared to the best competitor.

## CCS CONCEPTS

• **Information systems → Recommender systems**.

## KEYWORDS

bundle recommendation; curriculum learning; contrastive learning

**ACM Reference Format:**
Anonymous Author(s). 2024. Accurate Cold-start Bundle Recommendation via Popularity-based Coalescence and Curriculum Heating. In *Proceedings of Proceedings of The ACM Web Conference 2024 (The Web Conference '24).* ACM, New York, NY, USA, 9 pages. https://doi.org/XXXXXXX.XXXXXXX

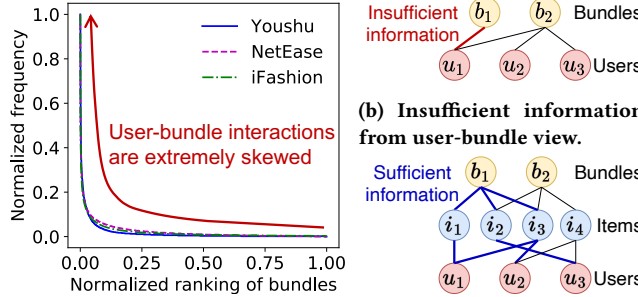

(a) Extremely skewed distribution of bundle interactions.

(b) **Insufficient information from user-bundle view.**

(c) **Sufficient information from user-item view.**

**Figure 1: (a) Extremely skewed distribution of bundle interactions in real-world datasets (data statistics are summarized in Table 1). (b-c) For an unpopular bundle, user-bundle view provides insufficient information while user-item view provides sufficient information.**

## 1 INTRODUCTION

*How can we accurately recommend cold-start bundles to users?* Bundle recommendation has garnered significant attention in both academia and industry since it enables providers to offer items to users with one-stop convenience [22]. In particular, recommending new bundles to users (i.e. cold-start bundle recommendation) has become crucial with the Web's evolution as new bundles are constantly created on the Web for various marketing purposes [10].

In recent years, bundle recommendation has seen advancements through matrix factorization-based approaches [4, 10, 25] and graph learning-based approaches [6, 14, 22]. However, they have been developed for a warm-start setting, where all bundles already possess historical interactions with users. Consequently, their efficacy diminishes in cold-start scenarios, where certain bundles are devoid of historical interactions. This is because warm-start methods rely highly on historical information of user-bundle interactions to discern collaborative signals between users and bundles.

On the other hand, the cold-start problem [28] in item recommendation has been extensively studied, with a focus on aligning behavior representations with content representations. For instance, generative methods have aimed to model the generation of item behavior representations using mean squared error [31, 36], metric learning [47], or adversarial loss [9]. Dropout-based methods [34, 51] have aimed to bolster robustness to behavior information by randomly dropping the behavior embedding in the training phase. More recently, contrastive learning-based methods [39, 50] have shown superior performance by reducing the discrepancy between the distributions of behavior and content information of items. However, the existing methods for cold-start item recommendation fail to achieve high performance in bundle recommendation because they lack the ability to effectively leverage the user-item historical interactions when representing bundles. Furthermore, none of the existing works have explicitly considered the skewed distribution of user-bundle interactions which is a pivotal aspect in bundle recommendation as shown in Figure 1a. For unpopular bundles, aligning behavior representations from insufficient historical information with content representations amplifies inherent biases and makes it difficult to learn meaningful representations; this results in sacrificing the performance on a warm-start setting to improve the performance on a cold-start setting (see Figure 2).

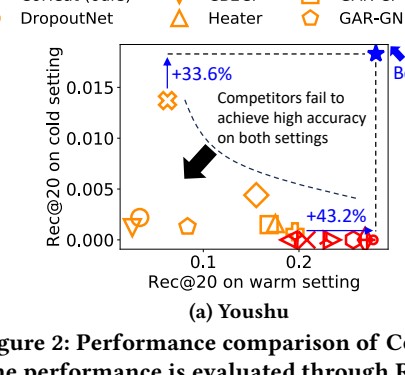 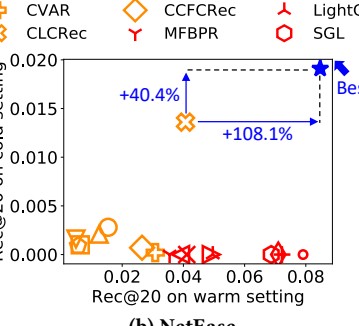 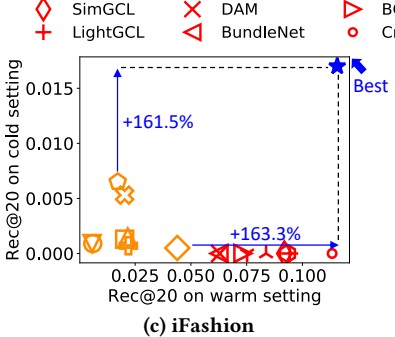

**Figure 2: Performance comparison of CoHeat with competitors on three real-world datasets: Youshu, NetEase, and iFashion. The performance is evaluated through Recall@20 for all experiments. We mark cold-start methods as orange, and warm-start methods as red. The cold-start methods typically sacrifice warm setting performance to excel in cold settings. The warm-start methods show poor performance in cold settings. CoHeat demonstrates superior performance over existing methods in both cold and warm settings, with a notable advantage in outperforming competitors.**

In this paper, we propose CoHeat (Popularity-based Coalescence and Curriculum Heating), an accurate method for cold-start bundle recommendation. CoHeat constructs representations of users and bundles using two distinct graph-based views: user-bundle view and user-item view. The user-bundle view is grounded in historical interactions between users and bundles, whereas the user-item view is rooted in bundle affiliations and historical interactions between users and items. To handle the extremely skewed distribution as shown in Figure 1a, CoHeat strategically leverages both views in its predictions, emphasizing user-item view for less popular bundles since they provide richer information than the sparse user-bundle view, as shown in Figures 1b and 1c. In addition, to effectively learn the user-item view representations which are fully used for cold-start bundles, CoHeat exploits a curriculum learning approach that gradually shifts the training focus from the user-bundle view to the user-item view. CoHeat further exploits a contrastive learning approach to align the representations of the two views effectively.

Our contributions are summarized as follows:

- **Problem.** To our knowledge, this is the first work that tackles the cold-start problem in bundle recommendation, a challenging problem of significant impact in real-world scenarios.
- **Method.** We propose CoHeat, an accurate method for cold-start bundle recommendation. CoHeat effectively learns user and bundle representations by considering the extremely skewed interactions to accurately recommend cold-start bundles based on their affiliations.
- **Experiments.** We experimentally show that CoHeat provides the state-of-the-art performance achieving up to 193% higher nDCG@20 compared to the best competitor in cold-start bundle recommendation while maintaining competitive performance in warm-start scenarios. (see Figure 2 and Table 2).

The rest of this paper is organized as follows. In section 2, we introduce the problem definition and preliminaries of CoHeat. We then propose CoHeat in Section 3, and present the experimental results in Section 4. We explain the related works in Section 5, and conclude in Section 6. The code and datasets are available at https://github.com/ColdBundle/CoHeat.

## 2 PRELIMINARIES

### 2.1 Problem Definition

The problem of cold-start bundle recommendation is defined as follows. Let $\mathcal{U}$, $\mathcal{B}$, and $\mathcal{I}$ be the sets of users, bundles, and items, respectively. Among the bundles, $\mathcal{B}_w \subset \mathcal{B}$ refers to the warm-start bundles that have at least one historical interaction with users, while $\mathcal{B}_c = \mathcal{B} \setminus \mathcal{B}_w$ represents the cold-start bundles that lack any historical interaction with users. The observed user-bundle interactions, user-item interactions, and bundle-item affiliations are defined respectively as $\mathcal{X} = \{(u,b)|u \in \mathcal{U}, b \in \mathcal{B}_w\}$, $\mathcal{Y} = \{(u,i)|u \in \mathcal{U}, i \in \mathcal{I}\}$, and $\mathcal{Z} = \{(b,i)|b \in \mathcal{B}, i \in \mathcal{I}\}$. Given $\{\mathcal{X}, \mathcal{Y}, \mathcal{Z}\}$, our goal is to recommend $k$ bundles from $\mathcal{B}$ to each user $u \in \mathcal{U}$. Note that the given interactions are observed only for warm bundles but the objective includes recommending also cold bundles to users.

The central challenge in cold-start bundle recommendation, compared to traditional bundle recommendation, lies in accurately predicting the relationship between a user $u \in \mathcal{U}$ and a cold-start bundle $b \in \mathcal{B}_c$ in the absence of any historical interactions of $b$. Hence, the crux of addressing the problem is to effectively estimate the representations of cold-start bundles using their affiliation information.

Bundles are compositions of multiple items, each having distinct interactions with users. This contrasts sharply with traditional cold-start item recommendations where contents are often represented as independent entities like texts or images. Moreover, user-bundle interactions exhibit a pronounced skewness, far more than typical user-item interactions. These make the cold-start bundle recommendation uniquely challenging compared to the cold-start item recommendation.

### 2.2 Curriculum Learning

Curriculum learning, inspired by human learning, structures training from simpler to more complex tasks, unlike standard approaches that randomize task order [2, 38]. Its effectiveness has been proven in various domains, including computer vision [44, 46], natural language processing [18, 45], robotics [21, 23], and recommender systems [8, 13].

In this work, we harness curriculum learning to enhance the learning process of user-bundle relationships. We initiate with a

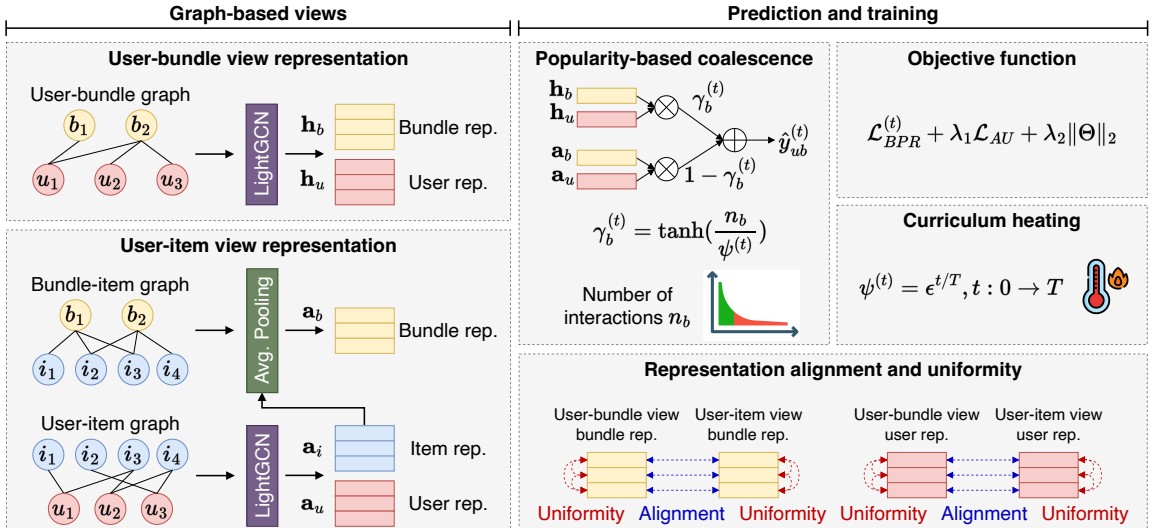

Figure 3: Overview of CoHeat (see Section 3 for details).

focus on the more straightforward user-bundle view embeddings and then progressively shift attention to the intricate user-item view embeddings. This strategy stems from the ease of learning user-bundle view embeddings, which directly capture collaborative signals from historical interactions. In contrast, user-item view embeddings are more complicated due to their dependence on the representations of affiliated items.

## 2.3 Contrastive Learning

Contrastive learning aims to derive meaningful embeddings by distinguishing between similar and dissimilar data samples. This approach has consistently demonstrated superior performance across a range of research fields, including computer vision [11, 27, 37], natural language processing [16, 42], and recommender systems [3, 35]. In bundle recommendation, CrossCBR [22] has utilized InfoNCE [32] as a contrastive learning function to regularize embeddings of users and bundles between the user-bundle and user-item views. However, its approach of aligning the two views without differentiation in prediction can be limiting, especially in cold-start scenarios where the user-bundle view is sparse.

In this work, we enhance the application of contrastive learning in bundle recommendation. Instead of treating the two views equally, we dynamically adjust their weights based on bundle popularity. This facilitates the transfer of information from a more informative view to the counterpart, enabling effective recommendations for both cold and warm bundles. Furthermore, we leverage the alignment and uniformity loss [37], which has been demonstrated to outperform InfoNCE in various applications [35, 37, 41]. This loss function directly optimizes the foundational principles of contrastive learning, ensuring more robust and meaningful embeddings.

## 3 PROPOSED METHOD

### 3.1 Overview

We address the following challenges to achieve high performance in cold-start bundle recommendation.

C1. **Handling highly skewed interactions.** Previous works depend overly on user-bundle view representations, which are unreliable if bundles have sparse interactions. How can we effectively learn the representations from highly skewed interactions?

C2. **Effectively learning user-item view representations.** Despite the ample information provided by the user-item view, multiple items in a bundle complicate the learning of these representations. How can we effectively learn the user-item view representations?

C3. **Bridging the gap between two view representations.** Aligning user-bundle and user-item views is crucial, as we estimate future interactions of cold bundles using only their affiliations. How can we effectively reconcile these two view representations?

To address these challenges, we propose CoHeat (Popularity-based Coalescence and Curriculum Heating) with the following main ideas.

I1. **Popularity-based coalescence.** For the score between users and bundles, we propose the coalescence of two view scores, with less popular bundles relying more on user-item view scores and less on user-bundle view scores.

I2. **Curriculum heating.** We propose a curriculum learning approach that focuses initially on training representations using the user-bundle view, gradually shifting the focus to the user-item view.

I3. **Representation alignment and uniformity.** We exploit a representation alignment and uniformity approach to effectively reconcile the user-bundle view and user-item view representations.

Figure 3 depicts the schematic illustration of CoHeat. Given user-bundle interactions, user-item interactions, and bundle-item affiliations, CoHeat forms two graph-based views. Then, it predicts user-bundle scores by coalescing scores from both views based on bundle popularity. During training, CoHeat prioritizes user-bundle view initially, transitioning progressively to user-item view via

## 3.2 Two Graph-based Views

The objective of bundle recommendation is to estimate the relationship between users and bundles by learning their latent representations. We utilize graph-based representations of users and bundles to fully exploit the given user-bundle interactions, user-item interactions, and bundle-item affiliations. We construct user-bundle view and user-item view graphs and use LightGCN [20] to obtain embeddings of users and bundles [22].

**User-bundle view representation and score.** In user-bundle view, we aim to capture the behavior signal between users and bundles. Specifically, we construct a bipartite graph using user-bundle interactions, and propagate the historical information using a LightGCN. The $k$-th layer of the LightGCN is computed as follows:

$$\mathbf{h}_u^{(k)} = \sum_{b \in \mathcal{N}_u} \frac{1}{\sqrt{|\mathcal{N}_u|}\sqrt{|\mathcal{N}_b|}} \mathbf{h}_b^{(k-1)}, \mathbf{h}_b^{(k)} = \sum_{u \in \mathcal{N}_b} \frac{1}{\sqrt{|\mathcal{N}_b|}\sqrt{|\mathcal{N}_u|}} \mathbf{h}_u^{(k-1)},$$
(1)

where $\mathbf{h}_u^{(k)}, \mathbf{h}_b^{(k)} \in \mathbb{R}^d$ are the embeddings of user $u$ and bundle $b$ at $k$-th layer, respectively; $\mathcal{N}_u$ and $\mathcal{N}_b$ are the sets of user $u$'s neighbors and bundle $b$'s neighbors in the user-bundle graph, respectively. $\mathbf{h}_u^{(0)}, \mathbf{h}_b^{(0)} \in \mathbb{R}^d$ are randomly initialized before the training of the model. We obtain the user-bundle view representations of user $u$ and bundle $b$ by aggregating the embeddings from all layers with a weighting approach that places greater emphasis on the lower layers as follows:

$$\mathbf{h}_u = \sum_{k=0}^{K} \frac{1}{k+1} \mathbf{h}_u^{(k)}, \mathbf{h}_b = \sum_{k=0}^{K} \frac{1}{k+1} \mathbf{h}_b^{(k)},$$
(2)

where $\mathbf{h}_u, \mathbf{h}_b \in \mathbb{R}^d$ are the user-bundle view embeddings of user $u$ and bundle $b$, respectively; $K$ denotes the last layer. Finally, the user-bundle view score between user $u$ and bundle $b$ is defined as $h_{ub} = \mathbf{h}_u^\top \mathbf{h}_b$.

**User-item view representation and score.** In user-item view, we aim to learn the relationship between users and bundles from the perspective of item affiliations. Specifically, we construct a bipartite graph using user-item interactions, and propagate the historical information using another LightGCN. Then, we obtain bundle representations by aggregating the affiliated items' representations. The $k$-th layer of the LightGCN is computed as follows:

$$\mathbf{a}_u^{(k)} = \sum_{i \in \mathcal{N}_u'} \frac{1}{\sqrt{|\mathcal{N}_u'|}\sqrt{|\mathcal{N}_i|}} \mathbf{a}_i^{(k-1)}, \mathbf{a}_i^{(k)} = \sum_{u \in \mathcal{N}_i} \frac{1}{\sqrt{|\mathcal{N}_i|}\sqrt{|\mathcal{N}_u'|}} \mathbf{a}_u^{(k-1)}, \quad (3)$$

where $\mathbf{a}_u^{(k)}, \mathbf{a}_i^{(k)} \in \mathbb{R}^d$ are the embeddings of user $u$ and item $i$ at $k$-th layer, respectively; $\mathcal{N}_u'$ and $\mathcal{N}_i$ are the sets of user $u$'s neighbors and item $i$'s neighbors in the user-item graph, respectively. $\mathbf{a}_u^{(0)}, \mathbf{a}_i^{(0)} \in \mathbb{R}^d$ are randomly initialized before the training. We obtain the user-item view representations of user $u$ and item $i$ by aggregating the embeddings from all layers with a weighting approach as follows:

$$\mathbf{a}_u = \sum_{k=0}^{K} \frac{1}{k+1} \mathbf{a}_u^{(k)}, \mathbf{a}_i = \sum_{k=0}^{K} \frac{1}{k+1} \mathbf{a}_i^{(k)},$$
(4)

where $\mathbf{a}_u, \mathbf{a}_i \in \mathbb{R}^d$ are the user-item view embeddings of user $u$ and item $i$, respectively; $K$ indicates the last layer. We then obtain the user-item view representations of bundle $b$ by an average pooling as $\mathbf{a}_b = \frac{1}{|\mathcal{N}_b'|} \sum_{i \in \mathcal{N}_b'} \mathbf{a}_i$, where $\mathcal{N}_b'$ is the set of bundle $b$'s affiliated items. Finally, the user-item view score between user $u$ and bundle $b$ is defined as $a_{ub} = \mathbf{a}_u^\top \mathbf{a}_b$.

## 3.3 Popularity-based Coalescence

For recommending bundles to users, our objective is to estimate the final score $\hat{y}_{ub} \in \mathbb{R}$ between user $u$ and bundle $b$ using scores $h_{ub}$ and $a_{ub}$, derived from the two distinct views. However, real-world datasets present an inherent challenge of handling the extremely skewed distribution of interactions between users and bundles, as illustrated in Figure 1a. While both views are informative, many unpopular bundles are underrepresented in the user-bundle view due to the insufficient interactions as illustrated in Figure 1b. In contrast, they are often sufficiently represented in the user-item view, as depicted in Figure 1c. A uniform weighting strategy for both views, as in CrossCBR, risks amplifying biases inherent to the user-bundle view, especially for the unpopular bundles. This predicament is further exacerbated for cold-start bundles devoid of interactions in user-bundle view.

To deal with this challenge, we propose two desired properties for the user-bundle relationship score $\hat{y}_{ub}$.

**Property 1** (User-bundle view influence mitigation): The influence of user-bundle view score should be mitigated as a bundle's interaction number decreases, i.e. $\frac{\partial \hat{y}_{ub}}{\partial h_{ub}} < \frac{\partial \hat{y}_{ub'}}{\partial h_{ub'}}$ if $n_b < n_{b'}$ where $n_b$ is the number of user interactions of bundle $b$.

**Property 2** (User-item view influence amplification): The influence of user-item view score should be amplified as a bundle's interaction number decreases, i.e. $\frac{\partial \hat{y}_{ub}}{\partial a_{ub}} > \frac{\partial \hat{y}_{ub'}}{\partial a_{ub'}}$ if $n_b < n_{b'}$ where $n_b$ is the number of user interactions of bundle $b$.

Properties 1 and 2 are crucial in achieving a balanced interplay between the user-bundle view and user-item view scores based on bundle popularities. Specifically, they ensure a heightened emphasis on the user-item view over the user-bundle view for less popular bundles.

We propose the user-bundle relationship score $\hat{y}_{ub}$ that satisfies the two desired properties by weighting the two scores $h_{ub}$ and $a_{ub}$ based on bundle popularities as follows:

$$\hat{y}_{ub} = \gamma_b h_{ub} + (1 - \gamma_b) a_{ub},$$
(5)

where $\gamma_b \in [0, 1]$, which is defined in the next subsection, denotes a weighting coefficient such that $\gamma_b > \gamma_{b'}$ if $n_b > n_{b'}$. A smaller value of $\gamma_b$ (i.e. a smaller value of $n_b$) ensures that the score $\hat{y}_{ub}$ is predominantly influenced by the user-item view score $a_{ub}$. We show in Lemmas 3.1 and 3.2 that Equation (5) satisfies all the desired properties.

LEMMA 3.1. *Equation* (5) *satisfies Property 1.*

PROOF. $\frac{\partial \hat{y}_{ub}}{\partial h_{ub}} = \gamma_b$. Thus, $\frac{\partial \hat{y}_{ub}}{\partial h_{ub}} < \frac{\partial \hat{y}_{ub'}}{\partial h_{ub'}}$ if $n_b < n_{b'}$ because $\gamma_b < \gamma_{b'}$. □

LEMMA 3.2. *Equation* (5) *satisfies Property 2.*

PROOF. $\frac{\partial \hat{y}_{ub}}{\partial a_{ub}} = 1 - \gamma_b$. Thus, $\frac{\partial \hat{y}_{ub}}{\partial a_{ub}} > \frac{\partial \hat{y}_{ub'}}{\partial a_{ub'}}$ if $n_b < n_{b'}$ because $1 - \gamma_b > 1 - \gamma_{b'}$. □

## 3.4 Curriculum Heating

Despite the ample information provided by the user-item view, multiple items in a bundle complicate the learning of user-item representations. This difficulty arises because accurate representation of a bundle necessitates well-represented embeddings of its all affiliated items, and each item further requires well-represented embeddings of the connected users. On the other side, the user-bundle view representation is relatively straightforward to learn. This simplicity arises because we encapsulate each bundle's historical characteristics into a single embedding rather than understanding the intricate composition of the bundle.

Hence, we propose to focus initially on learning easier view representations and gradually shift the focus to learning harder view representations. Thus, we modify Equation (5) by exploiting a curriculum learning approach that focuses initially on training user-bundle view representations, and gradually shifts the focus to the user-item view representations as follows:

$$\hat{y}_{ub}^{(t)} = \gamma_b^{(t)} h_{ub} + (1 - \gamma_b^{(t)}) a_{ub}, \tag{6}$$

where $\hat{y}_{ub}^{(t)} \in \mathbb{R}$ is the estimated relationship score between user $u$ and bundle $b$ at epoch $t$. $\gamma_b^{(t)} \in \mathbb{R}$ is defined as $\gamma_b^{(t)} = \tanh\left(\frac{n_b}{\psi^{(t)}}\right)$, where $n_b$ is the number of interactions of bundle $b$, and $\psi^{(t)} > 0$ is the temperature at epoch $t$. Note that $\gamma_b^{(t)}$ lies within the interval $[0, 1]$ because $\frac{n_b}{\psi^{(t)}} \geq 0$. Then, we incrementally raise the temperature $\psi^{(t)}$ up to the maximum temperature as follows:

$$\psi^{(t)} = \epsilon^{t/T}, t : 0 \to T, \tag{7}$$

where $t, T \in \mathbb{R}$ are the current and the maximum epochs of the training process, and $\epsilon > 1$ is the hyperparameter of the maximum temperature. In the initial epochs of training, $\gamma_b^{(t)}$ is large since $t$ is small. As a result, the score $\hat{y}_{ub}^{(t)}$ relies more heavily on $h_{ub}$ than $a_{ub}$. However, as the training progresses and $t$ increases, $\gamma_b^{(t)}$ diminishes, shifting the emphasis from $h_{ub}$ to $a_{ub}$. This heating mechanism is applied to all bundles regardless of their popularity. Furthermore, we show in Lemmas 3.3 and 3.4 that Equation (6) still satisfies the two desired properties.

Lemma 3.3. *Equation (6) satisfies Property 1.*

Proof. $\frac{\partial \hat{y}_{ub}^{(t)}}{\partial h_{ub}} = \tanh\left(\frac{n_b}{\psi^{(t)}}\right)$. Thus, $\frac{\partial \hat{y}_{ub}^{(t)}}{\partial h_{ub}} < \frac{\partial \hat{y}_{ub'}^{(t)}}{\partial h_{ub'}}$ if $n_b < n_{b'}$ because $\psi^{(t)}$ is the same for all bundles at epoch $t$ and $tanh(\cdot)$ is an increasing function. □

Lemma 3.4. *Equation (6) satisfies Property 2.*

Proof. $\frac{\partial \hat{y}_{ub}^{(t)}}{\partial a_{ub}} = 1 - \tanh\left(\frac{n_b}{\psi^{(t)}}\right)$. Thus, $\frac{\partial \hat{y}_{ub}^{(t)}}{\partial a_{ub}} > \frac{\partial \hat{y}_{ub'}^{(t)}}{\partial a_{ub'}}$ if $n_b < n_{b'}$ because $\psi^{(t)}$ is the same for all bundles at epoch $t$ and $1 - tanh(\cdot)$ is a decreasing function. □

## 3.5 Representation Alignment and Uniformity

While the user-bundle view and user-item view are crafted to capture distinct representations, aligning the two views is essential, especially when predicting future interactions of cold bundles solely based on user-item view representations. Moreover, aligning two views facilitates knowledge transfer between the two views. This is essential because we gradually change the learning focus of views by curriculum heating, and the alignment helps the success of curriculum heating by effectively transferring knowledge from a view with richer knowledge to the opposite view. To achieve this, we exploit a contrastive learning-based approach that reconciles the two views. Specifically, we use the alignment and uniformity loss [37] as a regularization for the representations of the two views. We firstly $l_2$-normalize the embeddings of the two views as follows:

$$\tilde{\mathbf{h}}_u = \frac{\mathbf{h}_u}{\|\mathbf{h}_u\|_2}, \tilde{\mathbf{a}}_u = \frac{\mathbf{a}_u}{\|\mathbf{a}_u\|_2}, \tilde{\mathbf{h}}_b = \frac{\mathbf{h}_b}{\|\mathbf{h}_b\|_2}, \tilde{\mathbf{a}}_b = \frac{\mathbf{a}_b}{\|\mathbf{a}_b\|_2}, \tag{8}$$

where $\mathbf{h}_u, \mathbf{h}_b \in \mathbb{R}^d$ are user-bundle view representations of user $u$ and bundle $b$, respectively; $\mathbf{a}_u, \mathbf{a}_b \in \mathbb{R}^d$ are user-item view representations of user $u$ and bundle $b$, respectively. Then, we define an alignment loss as follows:

$$l_{align} = \mathop{\mathbb{E}}_{u \sim p_{user}} \|\tilde{\mathbf{h}}_u - \tilde{\mathbf{a}}_u\|_2^2 + \mathop{\mathbb{E}}_{b \sim p_{bundle}} \|\tilde{\mathbf{h}}_b - \tilde{\mathbf{a}}_b\|_2^2, \tag{9}$$

where $p_{user}$ and $p_{bundle}$ are the distributions of users and bundles, respectively. The alignment loss makes the embeddings of the two views close to each other for each user and bundle. We also define a uniformity loss as follows:

$$l_{uniform} = \log \mathop{\mathbb{E}}_{u,u' \sim p_{user}} e^{-2\|\tilde{\mathbf{h}}_u - \tilde{\mathbf{h}}_{u'}\|_2^2} + \log \mathop{\mathbb{E}}_{u,u' \sim p_{user}} e^{-2\|\tilde{\mathbf{a}}_u - \tilde{\mathbf{a}}_{u'}\|_2^2}$$
$$+ \log \mathop{\mathbb{E}}_{b,b' \sim p_{bundle}} e^{-2\|\tilde{\mathbf{h}}_b - \tilde{\mathbf{h}}_{b'}\|_2^2} + \log \mathop{\mathbb{E}}_{b,b' \sim p_{bundle}} e^{-2\|\tilde{\mathbf{a}}_b - \tilde{\mathbf{a}}_{b'}\|_2^2}, \tag{10}$$

where $u'$ and $b'$ denote a user and a bundle distinct from $u$ and $b$, respectively. The uniformity loss ensures distinct representations for different users (or bundles) by scattering them across the space. Finally, we define the contrastive loss for the two views as follows:

$$\mathcal{L}_{AU} = l_{align} + l_{uniform}. \tag{11}$$

## 3.6 Objective Function and Training

To effectively learn the user-bundle relationship, we utilize Bayesian Personalize Ranking (BPR) loss [26], which is the most widely used loss owing to its powerfulness, as follows:

$$\mathcal{L}_{BPR}^{(t)} = \mathop{\mathbb{E}}_{(u,b^+,b^-) \sim p_{data}} - \ln \sigma(\hat{y}_{ub^+}^{(t)} - \hat{y}_{ub^-}^{(t)}), \tag{12}$$

where $p_{data}$ is the data distribution of user-bundle interactions, with $u$ denoting a user, $b^+$ indicating a positive bundle, and $b^-$ representing a negative bundle. We define the final objective function as follows:

$$\mathcal{L}^{(t)} = \mathcal{L}_{BPR}^{(t)} + \lambda_1 \mathcal{L}_{AU} + \lambda_2 \|\Theta\|_2, \tag{13}$$

where $\lambda_1, \lambda_2 \in \mathbb{R}$ are balancing hyperparameters for the terms, and $\Theta$ denotes trainable parameters of CoHeat. For the distributions $p_{user}$ and $p_{bundle}$, we use in-batch sampling which selects samples from the training batch of $p_{data}$ rather than the entire dataset. This approach has empirically demonstrated to mitigate the training bias in prior studies [35, 49]. All the parameters are optimized in an end-to-end manner through the optimization. We also adopt an edge dropout [22, 40] while training to enhance the performance robustness.

Table 1: Summary of three real-world datasets where "dens." denotes the density of a matrix.

| Dataset | Users | Bundles | Items | User-bundle (dens.) | User-item (dens.) | Bundle-item (dens.) | Avg. size of bundle |
|---|---|---|---|---|---|---|---|
| Youshu[1] | 8,039 | 4,771 | 32,770 | 51,377 (0.13%) | 138,515 (0.05%) | 176,667 (0.11%) | 37.03 |
| NetEase[1] | 18,528 | 22,864 | 123,628 | 302,303 (0.07%) | 1,128,065 (0.05%) | 1,778,838 (0.06%) | 77.80 |
| iFashion[1] | 53,897 | 27,694 | 42,563 | 1,679,708 (0.11%) | 2,290,645 (0.10%) | 106,916 (0.01%) | 3.86 |

[1] https://github.com/mysbupt/CrossCBR

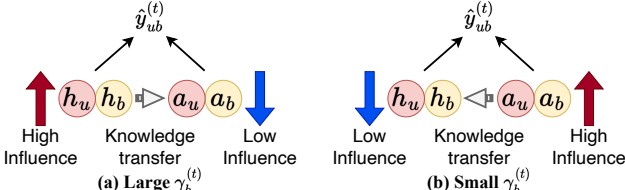

(a) Large $\gamma_b^{(t)}$    (b) Small $\gamma_b^{(t)}$

**Figure 4: Learning mechanism of CoHEAT (see Section 3.7 for details).**

## 3.7 Discussion of CoHEAT

The core of CoHEAT lies in its ability to dynamically adjust the weights of two distinct views, setting it apart from previous methods such as CrossCBR [22]. This dynamic adjustment is pivotal for achieving superior performance in the cold-start bundle recommendation.

Through the popularity-based coalescence, CoHEAT dynamically adjusts the weight $\gamma_b^{(t)}$ in Equation (6) to effectively harness the more informative view. For instance, when a bundle $b$ is popular, the influence of user-bundle view is bolstered with a large $\gamma_b^{(t)}$ because the bundle is rich of knowledge in this view. The knowledge then gets transferred to the user-item view by the alignment and uniformity loss, as depicted in Figure 4 (a). Conversely, for a less popular bundle, the influence of user-item view is amplified with a small $\gamma_b^{(t)}$, transferring the learned knowledge to the user-bundle view, as shown in Figure 4 (b). This strategy contrasts with CrossCBR, which may inadvertently amplify the underrepresented knowledge of unpopular bundles due to its uniform weighting strategy.

Additionally, the curriculum heating of CoHEAT further adjusts the weight $\gamma_b^{(t)}$ throughout the learning process. As the epochs progress, $\gamma_b^{(t)}$ diminishes (transitioning from Figure 4 (a) to Figure 4 (b)), ensuring both views are thoroughly utilized during training. This dynamic exchange of knowledge between two views results in CoHEAT's superior performance in both cold and warm settings, owing to the wealth of knowledge each view offers. This strategy is distinct from CrossCBR since the uniform weights for both views may lead to suboptimal results, especially in cold-start scenarios where the user-bundle view is sparse. Moreover, the curriculum heating strategically focuses on the easier view first, gradually shifting its attention to the more challenging view as the learning progresses. This helps a smoother and more effective knowledge transfer between the views.

## 4 EXPERIMENTS

In this section, we perform experiments to answer the following questions.

Q1. **Comparison with cold-start methods.** Does CoHEAT show superior performance in comparison to other cold-start methods in bundle recommendation?

Q2. **Comparison with warm-start methods.** Does CoHEAT show similar performance in warm-start bundle recommendation compared with baselines, although CoHEAT is a cold-start bundle recommendation method?

Q3. **Comparison by cold bundle ratio.** Does the performance difference between CoHEAT and baseline increase as the cold bundle ratio increases?

Q4. **Ablation study.** How do the main ideas of CoHEAT affect the performance?

Q5. **Effect of the maximum temperature.** How does the maximum temperature $\epsilon$, the critical hyperparameter, affect the performance of CoHEAT?

### 4.1 Experimental Setup

**Datasets.** We use three real-world bundle recommendation datasets as summarized in Table 1. Youshu [10] comprises bundles of books sourced from a book review site; NetEase [4] features bundles of music tracks from a cloud music service; iFashion [12] consists of bundles of fashion items from an outfit sales platform.

**Baseline cold-start methods.** We compare CoHEAT with existing cold-start item recommendation methods because they can be easily adapted for bundle recommendation by considering bundle-item affiliations as content information. DropoutNet [34] is a robustness-based method with a dropout operation. CB2CF [1] and Heater [51] are constraint-based methods that regularize the alignment. GAR [9] is a generative method with two variants GAR-CF and GAR-GNN. CVAR [48] is another generative method with a conditional decoder. CLCRec [39] and CCFCRec [50] are contrastive learning-based methods. We omit other competitors such as DUIF [17], MTPR [15], and NFM [19] because CLCRec and CCFCRec outperform them by a large margin on their extensive experiments. We also omit other generative competitors such as DeepMusic [31] and MetaEmb [24] because GAR [9] greatly outperforms them on its experiments. We use bundle-item multi-hot vectors as their content information.

**Baseline warm-start methods.** We also compare CoHEAT with previous warm-start recommendation methods. MFBPR [26] and LightGCN [20] are item recommendation methods with the modelings of matrix factorization and graph learning, respectively. SGL [40], SimGCL [43], and LightGCL [3] are the improved methods of item recommendation with contrastive learning approaches. DAM [10] is a bundle recommendation method with the modeling of matrix factorization. BundleNet [14], BGCN [6, 7], and Cross-CBR [22] are other bundle recommendation methods with the modeling of graph learning.

**Evaluation metrics.** We use Recall@$k$ and nDCG@$k$ metrics as in previous works [22, 39]. Recall@$k$ measures the proportion of relevant items in the top-$k$ list, while nDCG@$k$ weighs items by their rank. We set $k$ to 20. In tables, bold and underlined values indicate the best and second-best results, respectively.

**Table 2: Performance comparison of CoHeat and baseline cold-start methods on three real-world datasets.**

| Model | Youshu | | | | | | NetEase | | | | | | iFashion | | | | | |
|---|---|---|---|---|---|---|---|---|---|---|---|---|---|---|---|---|---|---|
| | Recall@20 | | | nDCG@20 | | | Recall@20 | | | nDCG@20 | | | Recall@20 | | | nDCG@20 | | |
| | Cold | Warm | All | Cold | Warm | All | Cold | Warm | All | Cold | Warm | All | Cold | Warm | All | Cold | Warm | All |
| DropoutNet [34] | .0022 | .0336 | .0148 | .0007 | .0153 | .0055 | .0028 | .0154 | .0046 | .0015 | .0078 | .0024 | .0009 | .0060 | .0039 | .0008 | .0045 | .0027 |
| CB2CF [1] | .0012 | .0258 | .0028 | .0007 | .0208 | .0021 | .0016 | .0049 | .0027 | .0006 | .0027 | .0014 | .0009 | .0057 | .0066 | .0006 | .0043 | .0048 |
| Heater [51] | .0016 | .1753 | .0541 | .0007 | .0826 | .0286 | .0021 | .0125 | .0102 | .0010 | .0064 | .0054 | .0015 | .0217 | .0123 | .0010 | .0151 | .0083 |
| GAR-CF [9] | .0015 | .1688 | .0529 | .0011 | .0726 | .0317 | .0010 | .0063 | .0014 | .0005 | .0035 | .0008 | .0013 | .0203 | .0090 | .0013 | .0143 | .0055 |
| GAR-GNN [9] | .0013 | .0835 | .0358 | .0006 | .0569 | .0178 | .0009 | .0056 | .0027 | .0003 | .0030 | .0012 | .0065 | .0172 | .0126 | .0030 | .0107 | .0087 |
| CVAR [48] | .0008 | .1958 | .0829 | .0002 | .1112 | .0533 | .0002 | .0308 | .0156 | .0001 | .0154 | .0084 | .0007 | .0220 | .0125 | .0004 | .0152 | .0084 |
| CLCRec [39] | .0137 | .0626 | .0367 | .0087 | .0317 | .0194 | .0136 | .0407 | .0259 | .0075 | .0215 | .0138 | .0053 | .0203 | .0126 | .0043 | .0135 | .0085 |
| CCFCRec [50] | .0044 | .1554 | .0702 | .0022 | .0798 | .0425 | .0007 | .0265 | .0130 | .0004 | .0128 | .0068 | .0005 | .0439 | .0252 | .0003 | .0304 | .0172 |
| CoHeat (ours) | .0183 | .2804 | .1247 | .0105 | .1646 | .0833 | .0191 | .0847 | .0453 | .0093 | .0455 | .0264 | .0170 | .1156 | .0658 | .0096 | .0876 | .0504 |

**Table 3: Performance comparison of CoHeat and baseline warm-start methods on three real-world datasets.**

| Model | Youshu | | NetEase | | iFashion | |
|---|---|---|---|---|---|---|
| | Recall@20 | nDCG@20 | Recall@20 | nDCG@20 | Recall@20 | nDCG@20 |
| MFBPR [26] | .1959 | .1117 | .0355 | .0181 | .0752 | .0542 |
| LightGCN [20] | .2286 | .1344 | .0496 | .0254 | .0837 | .0612 |
| SGL [40] | .2568 | .1527 | .0687 | .0368 | .0933 | .0690 |
| SimGCL [43] | .2691 | .1593 | .0710 | .0377 | .0919 | .0677 |
| LightGCL [3] | .2712 | .1607 | .0722 | .0388 | .0943 | .0686 |
| DAM [10] | .2082 | .1198 | .0411 | .0210 | .0629 | .0450 |
| BundleNet [14] | .1895 | .1125 | .0391 | .0201 | .0626 | .0447 |
| BGCN [6, 7] | .2347 | .1345 | .0491 | .0258 | .0733 | .0531 |
| CrossCBR [22] | .2776 | .1641 | .0791 | .0433 | .1133 | .0875 |
| CoHeat (ours) | .2804 | .1646 | .0847 | .0455 | .1156 | .0876 |

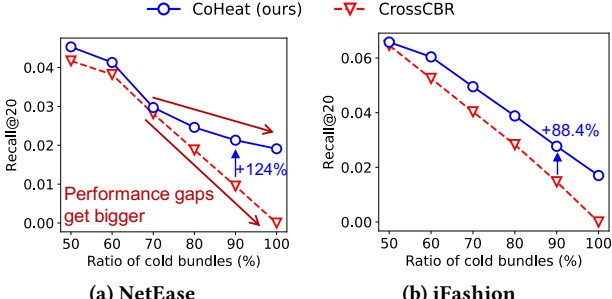

**Figure 5: Performance comparison by cold bundle ratio.**

**Experimental process.** We conduct experiments in warm-start, cold-start, and all-bundle scenarios as in previous works [39]. For the warm-start scenario, interactions are split into 7:1:2 subsets for training, validation, and testing. In the cold-start scenario, bundles are split in 7:1:2 ratio. In the all-bundle scenario, interactions are split in 7:1:2 ratio with a half for warm-start and the other half for cold-start bundles. We report the best Recall@20 and nDCG@20 within 100 epochs, averaged over three runs.

**Hyperparameters.** We utilize the baselines with their official implementations and use their reported best hyperparameters. We implement CoHeat with PyTorch. We set the dimensionality $d$ of node embeddings as 64. The other hyperparameters are grid-searched: the learning rate in {0.001, 0.0001, 0.00001}, $\lambda_1$ in {0.1, 0.2, 0.5, 1.0}, $\lambda_2$ in {0.00004, 0.0001, 0.0004, 0.001}, $K$ in {1, 2}, and the maximum temperature $\epsilon$ in {$10^1$, $10^2$, $10^3$, $10^4$, $10^5$, $10^6$}.

## 4.2 Comparison with Cold-start Methods (Q1)

In Table 2, we compare CoHeat with baseline cold-start methods. The results show that CoHeat consistently surpasses the baselines across all datasets and settings, verifying its superiority. Notably, CoHeat achieves 193% higher nDCG@20 compared to CCFCRec, the best competitor, on the iFashion dataset in the all-bundle scenario. The superiority of CoHeat over other cold-start methods stems from the following two key aspects. First, CoHeat adeptly harnesses collaborative information of each affiliated item in a bundle through the user-item view. This approach diverges from existing cold-start methods, which fall short in utilizing user-item interactions for bundle affiliations. Second, CoHeat explicitly addresses the pronounced skewness in user-bundle interactions through the proposed popularity-based coalescence. The results reveal the importance of tackling the inherent biases in distributions with extreme skewness, such as user-bundle interactions.

## 4.3 Comparison with Warm-start Methods (Q2)

Table 3 compares CoHeat with baseline warm-start methods in the warm-start scenario. Even though CoHeat is primarily designed for cold-start bundle recommendation, it surpasses all the baselines in the warm-start scenario as well. This indicates CoHeat effectively learns representations from both user-bundle and user-item views by dynamically adjusting the weights of two views in training. For the baselines, the performance improves when contrastive learning is used as exemplified in SGL, SimGCL, LightGCL, and CrossCBR. Additionally, graph-based models such as LightGCN, SGL, SimGCL, LightGCL, BGCN, and CrossCBR typically excel over other non-graph-based models. In light of these observations, Co-Heat strategically exploits a graph-based modeling approach and harnesses the power of contrastive learning. This makes CoHeat robustly achieve the highest performance across diverse scenarios.

## 4.4 Comparison by Cold Bundle Ratio (Q3)

In Figure 5, we compare the performance between CoHeat and CrossCBR on NetEase and iFashion datasets under varying cold bundle ratios in test datasets. We focus on investigating the performance disparity as conditions become increasingly colder, despite their analogous performance in warm settings, as shown in Table 3. The figure reveals a pronounced performance disparity between

**Table 4: Ablation study of CoHEAT in cold-start scenario which is our main target.**

| Model | Youshu | | NetEase | | iFashion | |
|---|---|---|---|---|---|---|
| | Recall @20 | nDCG @20 | Recall @20 | nDCG @20 | Recall @20 | nDCG @20 |
| CoHEAT-*PC* | .0000 | .0000 | .0000 | .0000 | .0000 | .0000 |
| CoHEAT-*CH-Ant* | .0177 | .0087 | .0176 | .0087 | .0164 | .0093 |
| CoHEAT-*CH-Fix* | .0180 | .0092 | .0182 | .0090 | .0164 | .0092 |
| CoHEAT-*AU* | .0069 | .0031 | .0029 | .0013 | .0013 | .0005 |
| **CoHEAT (ours)** | **.0183** | **.0105** | **.0191** | **.0093** | **.0170** | **.0096** |

CoHEAT and CrossCBR, intensifying as the cold bundle ratios increase. Remarkably, CrossCBR's performance plummets to zero in entirely cold conditions while CoHEAT maintains a more stable trajectory. This divergence is particularly accentuated in NetEase due to its sparser interactions and larger bundle size. The superiority of CoHEAT over CrossCBR is rooted in strategically reducing biases inherent in sparse interactions by adopting the popularity-based coalescence. Furthermore, CoHEAT enhances the learning of user-item view by exploiting curriculum heating, thereby utilizing bundle affiliation information more effectively. Thus, this approach is more beneficial for larger bundle sizes.

## 4.5 Ablation Study (Q4)

Table 4 provides an ablation study that compares CoHEAT with its four variants CoHEAT-*PC*, CoHEAT-*CH-Ant*, CoHEAT-*CH-Fix*, and CoHEAT-*AU*. This study is conducted in the cold-start scenario, which is the primary focus of our work. In CoHEAT-*PC*, we remove the influence of popularity-based coalescence by setting the value of $\gamma_b^{(t)}$ in Equation (6) to a constant 0.5. For CoHEAT-*CH-Ant*, we exploit an anti-curriculum learning strategy. The temperature in Equation (7) is defined as $t : T \rightarrow 0$, initiating the learning process with the user-item view and gradually shifting the focus to the user-bundle view. For CoHEAT-*CH-Fix*, we remove the effect of curriculum learning by setting the value of the $\psi^{(t)}$ in Equation (7) to the fixed maximum temperature $\epsilon$ regardless of epochs. For CoHEAT-*AU*, we omit $\mathcal{L}_{AU}$ from Equation (13), thereby excluding the contrastive learning between the two views. As shown in the table, CoHEAT consistently outperforms all the variants, which verifies all the main ideas help improve the performance. In particular, CoHEAT-*PC* shows a severe performance drop, justifying the importance of satisfying Properties 1 and 2 when addressing the extreme skewness inherent in cold-start bundle recommendation.

## 4.6 Effect of the Maximum Temperature (Q5)

The maximum temperature $\epsilon$ in Equation (7) is the most influential hyperparameter of CoHEAT since it directly affects both popularity-based coalescence and curriculum heating. Accordingly, we analyze the influence of $\epsilon$ in cold-start scenario on real-world datasets, as depicted in Figure 6. As shown in the figure, CoHEAT shows low performance for the extreme low temperature because the representations of user-item view are not sufficiently learned. For the extreme high temperature, the performance degrades because the speed of the curriculum is too fast to fully learn the representation of the two views. As a result, we set $\epsilon$ to $10^4$ for all datasets since it shows the best performance.

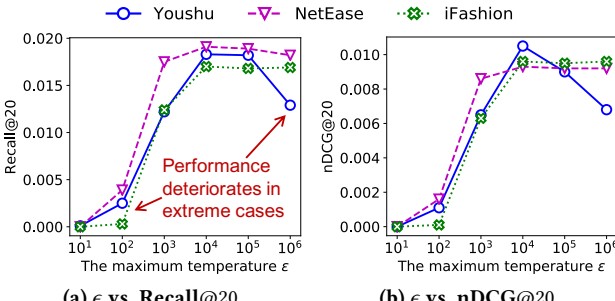

(a) $\epsilon$ vs. Recall@20      (b) $\epsilon$ vs. nDCG@20

**Figure 6: Effect of the maximum temperature $\epsilon$.**

## 5 RELATED WORKS

**Bundle recommendation.** Our work focuses on the cold-start problem in bundle recommendation. Previous works can be categorized based on their modeling structures: matrix factorization-based models [4, 10, 25] and graph learning-based models [6, 7, 14, 22]. Such methods operate under the assumption that all bundles have historical interactions, which makes them ill-suited for tackling the cold-start problem. However, in real-world scenarios, new bundles are introduced daily, leading to an inherent cold-start challenge. Our work addresses this significant yet overlooked issue, recognizing its potential impact on the field.

**Cold-start recommendation.** The cold-start problem, a long-standing challenge in recommender systems, focuses on recommending cold-start items with which users have not yet interacted. Existing works are primarily categorized into content-based methods [36, 47], generative methods [5, 9, 30, 31, 48], dropout-based methods [15, 29, 34], meta-learning methods [24, 33], and constraint-based methods [1, 39, 50, 51]. However, these methods are designed for item recommendation where contents are often represented as independent entities such as bag-of-words vectors, texts, or images. Moreover, such prior works have not explicitly addressed the highly skewed distribution of interactions, a critical aspect in bundle recommendation. Thus, our work excels over these methods in cold-start bundle recommendation by effectively harnessing intricate bundle affiliations from the user-item view and addressing the skewed distribution during training.

## 6 CONCLUSION

We propose CoHEAT, an accurate method for cold-start bundle recommendation. CoHEAT strategically leverages user-bundle and user-item views to handle the extremely skewed distribution of bundle interactions. By emphasizing the user-item view for less popular bundles, CoHEAT effectively captures richer information than the often sparse user-bundle view. The incorporation of curriculum learning further enhances the learning process, starting with the simpler user-bundle view embeddings and gradually transitioning to the more intricate user-item view embeddings. In addition, the contrastive learning of CoHEAT bolsters the learning of representations from the two views facilitating effective knowledge transfer from the richer to the sparser view. Extensive experiments show that CoHEAT provides the state-of-the-art performance in cold-start bundle recommendation, achieving up to 193% higher nDCG@20 compared to the best competitor.

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
