# OpenReview forum: "Accurate Cold-start Bundle Recommendation via Popularity-based Coalescence and Curriculum Heating"
_ACM.org/TheWebConf/2024/Conference — TheWebConf24 Oral_

### Official Review · Reviewer_GBKS · 2023-11-23

**Novelty:** 4
**Technical Quality:** 5

**Review:**

This paper works on solving the cold-start bundle recommendation problem. Traditional methods for cold-start item recommendation are not directly applicable to bundles as they overly depend on historical data, even for less popular bundles, failing to address the skewed distribution of bundle interactions. The authors propose CoHeat, which incorporates both the user-bundle graph and user-item graph to learn a better representation of the user and bundle. Authors utilize popularity-based coalescence which combines scores from two views with less popular bundles relying more on user-item view scores. What’s more, it utilizes curriculum learning and contrastive learning to balance and align the user and bundle representation in both two views. The authors also conduct a comprehensive experiment to validate the performance of CoHeat in both the cold setting and warm setting.

Pros:

a.	The writing is easy to follow, and the overview of CoHeat is very clear and pithy.

b.	The problem is new and important.

c.	The authors have conducted comprehensive experiments to evaluate the performance of CoHeat, and the improvement is significant compared with the baselines in both the cold setting and warm setting.

Cons:

a. It is necessary to compare two SOTA baselines on the warm setting.

    1. Distillation-Enhanced Graph Masked Autoencoders for Bundle Recommendation. Ren et al. SIGIR 2023

    2. Strategy-aware Bundle Recommender System. Wei et al. SIGIR 2023

b. The experiments are not very sufficient. It needs more exploration for the key modules. This additional analysis will provide a more comprehensive evaluation.

c. It should include more discussion about the more cold-start scenarios, such as cold-start items.

d. It's unclear how well CoHeat generalizes to different types of bundles or scales with larger datasets.

**Questions:**

Can this work adapt some cold-start recommender models as baselines since this work focuses on the cold-start bundle recommendation?

**Reviewer Confidence:**

3: The reviewer is confident but not certain that the evaluation is correct

**Scope:**

3: The work is somewhat relevant to the Web and to the track, and is of narrow interest to a sub-community

---

### Official Review · Reviewer_pApV · 2023-11-24

**Novelty:** 5
**Technical Quality:** 4

**Review:**

This paper introduces the CoHeat model for addressing the cold-start bundle recommendation problem, a significant challenge in practical applications. The model captures collaborative information through graphical views and incorporates a popularity-based coalescence and curriculum heating strategy to counteract the extreme skewness in data distribution.

Pros:
- The paper provides a comprehensive study of the extreme cold-start problem in bundle recommendation, utilizing a curriculum heating strategy to guide knowledge learning across different views.
- The theoretical explanations are compelling, and the inclusion of GitHub links for code sharing enhances the paper's reproducibility.
- The article is well-structured, focused, and engaging.

Cons:

- The authors claim to be the first to address the cold-start problem in bundle recommendation. However, existing methods, such as the one detailed in [1], have already explored this issue in both bundle and group recommendations. To strengthen its persuasiveness, the paper should include relevant background information, delineate its distinctions from the referenced paper [1], and compare methodologies in the experimental section.

    [1] Addressing the Extreme Cold-Start Problem in Group Recommendation [J]. arXiv preprint arXiv:2210.09672, 2022.

- The description of the method's innovation seems to overlook its core novelty. Unless I have overlooked key information, the primary innovation in this paper appears to be the application of a curriculum heating strategy to guide learning in both views. The emphasis on "effectively learning user and bundle representations by considering extremely skewed interactions for accurately recommending cold-start bundles based on their affiliations" has been addressed in previous methods.

- Figure 3 presents various types of graphs that overlap, making it challenging to distinguish between different methods. A simplification of the graphical representation would be beneficial for clarity.

**Questions:**

Please refer to the Cons in the review.

**Reviewer Confidence:**

4: The reviewer is certain that the evaluation is correct and very familiar with the relevant literature

**Scope:**

3: The work is somewhat relevant to the Web and to the track, and is of narrow interest to a sub-community

---

### Official Review · Reviewer_Z4bs · 2023-11-25

**Novelty:** 5
**Technical Quality:** 5

**Review:**

This paper presents an elegant solution for recommending cold bundles by leveraging user and bundle representations learned based on bundle affiliations as well as user-bundle interactions. The work coalesces the user-item and user-bundle scores by using popularity of the bundles as a factor. To warm start the problem, the authors suggest doing a curriculum learning on the easier problem first (user-bundle interactions) and gradually moving to user-item learning with more epochs.

The paper is really well written and well motivated with ideas and desired properties, and solutions are proposed based on that. While the ideas are elegant, they don't seem very novel to use this type of coalescing and curriculum based approaches. Nevertheless, the experiments are thorough and give me a lot of confidence in this elegant work.

**Questions:**

1) While certain types of curriculum learning do make sense, but I am not fully convinced on the authors' claim on user-bundles being easier to learn. It is true in terms of the amount of data, but there are a lot more nuances in bundles than individual items - e.g., a user may be interested in 90% of the items in a bundle and may consume that bundle repeatedly but trying to learn from that may skew the model to miss the 10% of the user interests early. Not sure if that's purposefully designed or I misunderstood something.

2) The dependence on popularity based coalescing seems fairly simplistic - the current formulation suggests that niche bundles would almost entirely be item dependent, whereas popular items would be bundle dependent. This seems fine in the general case, but the n_b in the tanh function should likely need to be normalized on a bundle's characteristics. There would be certain bundles that aren't meant to be globally popular but can still generate enough hyper active signals to entirely rely on bundle characteristics (imagine - niche video game bundles, or specific event playlists etc.)

**Reviewer Confidence:**

3: The reviewer is confident but not certain that the evaluation is correct

**Scope:**

4: The work is relevant to the Web and to the track, and is of broad interest to the community

---

### Official Review · Reviewer_P1nX · 2023-11-26

**Novelty:** 5
**Technical Quality:** 5

**Review:**

This paper presents a recommendation method for cold-start bundle  recommendation. The proposed method combines user-bundle view and user-item view, and also adopts curriculum learning and contrastive learning. Experimental results show the effectiveness of the proposed method.

Pros:
- The paper is generally well-written.
- The popularity-based coalescence is reasonable.
- The experiment improvements are significant.

Cons:
- One of the key insights of the proposed method is the popularity-based coalescence of the two views, right? This is mentioned in the title and the abstract, but not mentioned in the introduction.
- The lemmas in Section 3 are trivial.
- The first row (CoHear-PC) of Table 4 is all zeros. Is this a typo? If not, what are the reasons?

**Questions:**

- Why not mention popularity-based coalescence in intro?
- Why is the first row (CoHear-PC) of Table 4 all zeros?

**Reviewer Confidence:**

3: The reviewer is confident but not certain that the evaluation is correct

**Scope:**

3: The work is somewhat relevant to the Web and to the track, and is of narrow interest to a sub-community

---

### Official Review · Reviewer_UhEM · 2023-11-28

**Novelty:** 4
**Technical Quality:** 5

**Review:**

Summary

This paper proposes a novel method for cold-start bundle recommendation by considering bundle popularity, where curriculum heating and contrastive learning are exploited to implement the solution. Experiments on three real-world datasets demonstrate the effectiveness of the proposed method.

Positive Points

P1. The paper is well-written and easy to follow.

P2. The research question, cold-start issues in bundle recommendation is practical and useful in both academia and industry.

P3. The proposed method makes sense and works well in real-world datasets.


Negative Points


N1. Based on the definition in Section 2.1., cold start bundles refer to  “bundles that lack any historical interaction with users”. If the user (bundle) has no interaction with any bundles (users), how to measure the accuracy in the final?

N2. Some important related works are missing. For instance,

Personalized bundle list recommendation. WWW 2019

Consistency-aware recommendation for user-generated item list continuation. WSDM, 2020.

BRUCE: bundle recommendation using contextualized item embeddings. RecSys, 2022
Bundle MCR: towards conversational bundle recommendation. RecSys, 2022

Build your own bundle - a neural combinatorial optimization method. MM, 2021
A hierarchical self-attentive model for recommending user-generated item lists. CIKM, 2019
Suger: A subgraph-based graph convolutional network method for bundle recommendation. CIKM, 2022

Revisiting Bundle Recommendation: Datasets, Tasks, Challenges and Opportunities for Intent-aware Product Bundling. SIGIR, 2022.
Towards personalized bundle creative generation with contrastive non-
autoregressive decoding. SIGIR, 2022.
Strategy-aware bundle recommender system. SIGIR, 2023
Distillation-enhanced graph masked autoencoders for bundle recommendation. SIGIR, 2023

Multi-view intent disentangles graph networks for bundle recommendation. AAAI, 2022
Unifying multi-associations through hypergraph for bundle recommendation. KBS, 2022

N3. The proof or more explanations for the correctness and validity of the two properties may need to be provided.


N4. The proposed method is evaluated based on nDCG@20. I believe nDCG@5/10 should be considered. This is mainly because users may care more about the top-ranked products in the recommendation list. Does the proposed method outperform baselines regarding these scenarios? And to what extent does it outperform baselines?

**Questions:**

Please refer to the negative points.

**Reviewer Confidence:**

4: The reviewer is certain that the evaluation is correct and very familiar with the relevant literature

**Scope:**

3: The work is somewhat relevant to the Web and to the track, and is of narrow interest to a sub-community

---

### Official Review · Reviewer_A6nE · 2023-11-29

**Novelty:** 5
**Technical Quality:** 5

**Review:**

**Summary**

The paper presents CoHeat (Popularity-based Coalescence and Curriculum Heating) method for cold-start bundle recommendation. It constructs representations of users and bundles with user-bundle and user-item views with the former handles historical interactions between users and bundles while the later tackles bundle affiliations and historical interactions between users and items. Authors argue that CoHeat handles skewed distribution by leveraging both views in its predictions, emphasizing user-item view for less popular bundles. Extensive experiments showed significant improvements over baseline methods on three datasets.

**Strengths**

The reviewer believes that the following points represent the strength of the paper:

Overall the paper is well written, and motivated. Results indicate significant improvements over baseline methods in cold start bundle scenarios.

**Weakness**

The reviewer believes that the following points needs further explanation:

--- Authors listed a number of methods in cold and warm-start settings. However, the choices are not well motivated. The significant improvements indicate they might not be suitable. The reviewer appreciate if more details or motivation maybe provided on the choice of the baseline methods.

--- The reviewer is unable to pinpoint components which are responsible for significant performance boost over baseline methods.

**Questions:**

**Some other minor points**

1. I would recommend authors to add a reference for L87-91.


2. To include **accurate** in the title and L135 is unnecessary

3. Fig. 2 contain too much information and it requires reading of experimental section to understand it. Therefore, review suggest to improve the presentation or trim down the information. Though, the reviewer believe that it might be subjective and it might be useful for other readers.

4. The reviewer has also minor concern regarding novelty as the approach is using existing methods. I would appreciate if authors can list down their contributions in using contrastive learning, curriculum learning and the loss formulation.

**Reviewer Confidence:**

2: The reviewer is willing to defend the evaluation, but it is likely that the reviewer did not understand parts of the paper

**Scope:**

4: The work is relevant to the Web and to the track, and is of broad interest to the community

---

### Decision · Program_Chairs · 2024-01-22

**Decision:**

Accept (Oral)

**Comment:**

Quality:
 + Well-written paper
 + A real-world use case is addressed.
 + Methodology is sound
 - Quite many related references are missing.
 - Some minor comments have been raised, e.g., choice of 20 as cutoff for NDCG computation. They can be easily addressed.
 - Significant improvements over baseline.

 Clarity:
 + Writing is clear and easy to follow.

 Originality:
 + While existing methods are used in the components of the proposed approach, authors in their rebuttal rather convincingly states the original contributions.
 - However, authors do not mention all relevant works. So, while their approach has some novel components (which seem enough to warrant publication), authors need to include more relevant background information and link their approach better to existing works.

 Significance:
 + The work has some significance for WEB, in particular for the recommender systmems community.
 - Bundle recommendation itself is a rather niche topic.


 Side note: I also think "accurate" should be dropped from the title. And also this term should be generally used more sparingly in the paper itself. We anyway assume that the proposed method is more accurate w.r.t. some accuray-related measure than the state of the art. So, no need to mention it that many times.